# Elderly Population with COVID-19 and the Accuracy of Clinical Scales and D-Dimer for Pulmonary Embolism: The OCTA-COVID Study

**DOI:** 10.3390/jcm10225433

**Published:** 2021-11-20

**Authors:** Maribel Quezada-Feijoo, Mónica Ramos, Isabel Lozano-Montoya, Mónica Sarró, Verónica Cabo Muiños, Rocío Ayala, Francisco J. Gómez-Pavón, Rocío Toro

**Affiliations:** 1Cardiology Departament, Hospital Central de la Cruz Roja, C/Reina Victoria, 24, 28003 Madrid, Spain; monica.ramos81@gmail.com (M.R.); rayalamunoz@gmail.com (R.A.); 2Facultad de Medicina, Universidad Alfonso X El Sabio, Avda. De la Universidad, 1, Villanueva de la Cañada, 28691 Madrid, Spain; isalozanomontoya@hotmail.com (I.L.-M.); javiergomezpav@gmail.com (F.J.G.-P.); 3Geriatric Departament, Hospital Central de la Cruz Roja, C/Reina Victoria, 24, 28003 Madrid, Spain; 4Radiology Departament, Hospital Central de La Cruz Roja, C/Reina Victoria, 24, 28003 Madrid, Spain; monicasarrorx@telefonica.net; 5Biochemistry Laboratory, Hospital Central de la Cruz Roja, C/Reina Victoria, 24, 28003 Madrid, Spain; veronica.cabo@salud.madrid.org; 6Biomedical Research and Innovation Institute of Cádiz (INiBICA), Research Unit, Puerta del Mar University Hospital, Av/Ana de Viya 21, 11009 Cádiz, Spain; rociotorogreen@gmail.com; 7Medicine Department, School of Medicine, Cádiz University, Edificio Andrés Segovia 30 Floor, C/Dr Marañón S/N, 21001 Cádiz, Spain

**Keywords:** pulmonary embolism, Wells scale, Geneva scale, COVID-19, older patients

## Abstract

Background: Elderly COVID-19 patients have a high risk of pulmonary embolism (PE), but factors that predict PE are unknown in this population. This study assessed the Wells and revised Geneva scoring systems as predictors of PE and their relationships with D-dimer (DD) in this population. Methods: This was a longitudinal, observational study that included patients ≥75 years old with COVID-19 and suspected PE. The performances of the Wells score, revised Geneva score and DD levels were assessed. The combinations of the DD level and the clinical scales were evaluated using positive rules for higher specificity. Results: Among 305 patients included in the OCTA-COVID study cohort, 50 had suspected PE based on computed tomography pulmonary arteriography (CTPA), and the prevalence was 5.6%. The frequencies of PE in the low-, intermediate- and high-probability categories were 5.9%, 88.2% and 5.9% for the Geneva model and 35.3%, 58.8% and 5.9% for the Wells model, respectively. The DD median was higher in the PE group (4.33 mg/L; interquartile range (IQR) 2.40–7.17) than in the no PE group (1.39 mg/L; IQR 1.01–2.75) (*p* < 0.001). The area under the curve (AUC) for DD was 0.789 (0.652–0.927). After changing the cutoff point for DD to 4.33 mg/L, the specificity increased from 42.5% to 93.9%. Conclusions: The cutoff point DD > 4.33 mg/L has an increased specificity, which can discriminate false positives. The addition of the DD and the clinical probability scales increases the specificity and negative predictive value, which helps to avoid unnecessary invasive tests in this population.

## 1. Introduction

The diagnosis of pulmonary embolism (PE) in older patients is challenging due to the associated comorbidity [1] and masked symptoms [2]. An increased risk of PE, which is a medical emergency with a high mortality rate, and the development of a hypercoagulable state have been widely known to be associated with severe acute respiratory syndrome coronavirus 2 (SARS-CoV-2) disease 2019 (COVID-19) [3,4,5].

The preliminary diagnosis of PE begins with clinical suspicion established by a pretest probability using diagnostic scoring systems such as the Wells criteria or the Geneva scoring criteria [6,7]. These scales have been previously validated and may be useful to identify patients who have a low and intermediate probability of developing a PE.

To date, scales have been validated with a normal D-dimer (DD) level; however, this assessment is limited within the elderly population because as the age of the patient increases, the DD specificity decreases. Additionally, the plasma levels of DD increase as part of COVID-19 infection, making the use of these scales difficult. DD has been proven to be a significant predictor of increased mortality in patients infected by COVID-19 [8,9] but is not specific for the diagnosis of PE. There are also other markers, such as troponin and N-terminal pro-brain natriuretic peptide (NT-pro-BNP), that are increased in COVID-19 patients [10] and are also related to a worse prognosis [11]. For this reason, it is crucial to determine the cutoff level for DD in this population to improve the diagnosis of PE in frail and elderly populations, and this can add diagnostic value to the clinical scales for the requirements of tailoring imaging tests in highly suspicious PE cases [12].

We sought to explore the diagnostic accuracy and reproducibility of the Wells and Geneva clinical probability scales and their association with DD in the diagnosis of PE in elderly patients with COVID-19.

## 2. Materials and Methods

### 2.1. Study Population

A longitudinal, observational study was designed. Patients over 75 years of age hospitalized with COVID-19 with a clinical suspicion of PE were recruited from the Acute Geriatrics Unit between March and May 2020. This study belongs to the OCTA-COVID-19 cohort. Patients under 75 years of age, those with palliative needs, those diagnosed by the attending team and those who did not meet the diagnostic criteria for COVID-19 were excluded. Patients with a high suspicion of PE who could not undergo a computed tomography (CT) scan and those who declined to participate were also excluded.

### 2.2. Ethics Approvals

The study protocol was approved by the ethics committee (PI-4134) and was conducted in full compliance with the Declaration of Helsinki. All participants provided written informed consent. The collected data were appropriately made anonymous, and each patient was identified by a unique alphanumeric identification code.

### 2.3. Assessment of Clinical Probability

The Wells and revised Geneva scores were calculated to evaluate the probability of PE. Recruited patients were classified based on both scores into one of three categories. Based on the Wells scale [13,14], we considered low risk to be less than 2 points, moderate risk from 2 to 6 points and high risk over 6 points. Meanwhile, the Geneva scale [15] was scored as low probability for PE under 3 points, intermediate from 4 to 10 points and high probability over 11 points. A positive computed tomography pulmonary arteriography (CTPA) confirmed the presence of PE. Patients with a decreased renal filtration <30 mL/mL/min/1.73 m^2^ did not undergo the CT scan. In patients with renal filtration between 30 and 45 mL/min, a hydration protocol was carried out prior to the scan (Figure 1).

We considered terminal criteria as advanced organ disease, such as severe lung failure with irreversible damage, end-stage heart disease, advanced kidney disease without dialysis criteria or advanced treatment subsidiary, among others, according to The National Hospice Organization.

### 2.4. Data Collection

The biodemographic data and clinical characteristics were collected prospectively from patients with a clinical suspicion of PE. The clinical signs that were assessed included heart rate, breathing rate, oxygen saturation, pain in the deep vein of the lower limb during palpation and unilateral edema. The risk factors that were considered included atrial fibrillation, deep vein thrombosis (DVT) or PE, cancer, bed rest for more than 3 days, newly confirmed DVT events and the presence of associated arterial ischemia. Data regarding clinical complications were collected during hospitalization. The CURB-65 scale, as recommended by the British Thoracic Society [16], was applied to stratify the severity of pneumonia. The degree of frailty was assessed by the Clinical Frailty Scale (CFS) two weeks prior to hospitalization [17]. The degree of dependence was calculated using the Barthel Scale [18], and the presence of dementia was assessed by the Global Deterioration Scale (GDS) [19].

### 2.5. Laboratory Procedures

SARS-CoV-2 detection was performed using real-time reverse transcriptase-polymerase chain reaction on nasal swabs. The routine blood examinations that were performed included a complete blood count and serum biochemical tests. The determination of plasmatic DD was carried out using the VIDAS^®^ DD technique (bioMérieux, Lyon, France), a sandwich-type immuno-enzymatic method in 2 stages, with a final detection by the enzyme-linked fluorescence assay (ELFA). The concentration was expressed in micrograms per milliliter of fibrinogen equivalent units. We included the peak DD value, either from the beginning of admission or during the course of hospitalization. Cardiac biomarkers were measured in patients who were suspected of having cardiac involvement. NT-ProBNP was analyzed using a VIDAS analyzer (bioMérieux, Lyon, France). Troponin T (TnT) levels were measured using a (DADE Stratus^®^, Roche Diagnostics, Switzerland), fluorometric enzyme immunoassay analyzer with specific reagents and calibrators.

### 2.6. Definitions

The DD value was adjusted based on the patient’s age following the current PE guidelines [20] and was considered elevated when it was above 1 mg/L. NT-proBNP was considered elevated when it was above 450 pg/mL in patients with a normal sinus rhythm and when it was above 1100 pg/mL in patients with atrial fibrillation [21]. Acute myocardial damage was considered in patients with TnT levels above the 99th percentile of the upper limit of normal (50 ng/L) [22].

### 2.7. Statistical Analysis

Continuous variables are summarized as the median and interquartile range (IQR), and categorical data are summarized as frequencies and percentages. For comparisons, the Mann–Whitney U test was used because of the non-normal distribution of the continuous data. Categorical data were compared using the chi-square test or Fisher’s test, based on the expected counts. Ordinal data were compared with the Cochran-Armitage trend test.

The performances of the Wells score, the revised Geneva score and DD as diagnostic predictors of PE were assessed. The area under the receiver operating characteristic (ROC) curve was estimated, and a cutoff value was calculated using Youden’s index. The sensitivity, specificity and predictive values and their 95% confidence limits were calculated. The combinations of DD and the clinical scales were performed using both positive rules for higher specificity. All analyses were performed with SAS^®^ 9.4 (SAS Institute Inc., Cary, NC, USA). A *p*-value ≤ 0.05 was considered statistically significant.

## 3. Results

### 3.1. Participant Characteristics

The OCTA-COVID study included 305 patients admitted for COVID-19 pneumonia, and 50 patients who had suspected PE based on CTPA were included in our study. Seventeen patients were confirmed to have a diagnosis of PE, and the mean hospitalization stay was 14.5 days (IQR 11–21). Thus, the prevalence of PE in the global population was 5.6%. In our cohort, 70% of the patients were previously treated with heparin prophylaxis, and 24% were treated with full doses of anticoagulation. Regarding the distribution of PE, 11 patients (64.7%) had peripheral PE, 5 patients (29.4%) had central and peripheral PE and only 1 patient (5.9%) had central PE. Segmental artery involvement was seen in 8 patients (50%), subsegmental arteries were seen in 2 patients (12%) and 37.5% of the patients had both segmental and subsegmental involvement.

The anthropometric, clinical and geriatric characteristics of all patients with suspected PE related to COVID-19 infection are shown in Table 1. 

While the patients with PE were more obese (*p* = 0.046), no difference was noted as related to clinical antecedents, such as previous PE, DVT, trauma or palliative cancer treatment. Previous oncological diseases were noted in 35.3% of the patients with PE, with a trend toward statistical significance between the groups (*p* = 0.07).

Based on their clinical situation, patients diagnosed with PE presented with a significantly higher heart rate of 96 beats per minute than non-PE patients (*p* = 0.015). Asthenia was more prevalent as a clinical symptom in patients without PE, with a statistically significant difference (*p* = 0.052).

Regarding the biomarkers, the DD median was higher in the PE group, with a statistically significant difference (4.3 mg/L; IQR 2.40–7.17 vs. 1.3 mg/L; IQR 1.01–2.75; *p* < 0.001) Table 2.

Considering the biomarker evolution during hospitalization in our elderly cohort, the DD level remained elevated until the fifth day of infection in patients who presented with PE, while the non-PE cohort showed a gradually decreased DD level. A high value remained in the PE group on the third day, with a statistically significant difference from the group without PE (3.5 vs. 1.2 mg/L; *p* < 0.006). Likewise, the C-reactive protein (CRP) level was increased in the PE group on the third day but decreased gradually starting on the first day in the non-PE group (Figure 2a,b).

### 3.2. Predictive Values of the Wells Score and the Revised Geneva Score in the Elderly Group with COVID-19 and Pulmonary Embolism

The frequencies of PE in the low-, intermediate- and high-probability categories were 5.9%, 88.2% and 5.9% for the Geneva model and 35.3%, 58.8% and 5.9% for the Wells scale, respectively. The average score of the Wells scale in the PE cohort was higher than that in the group without PE (3 (IQR 1.5–3) vs. 1.5 (1.5–2.5); *p* = 0.06), but the difference was not statistically significant. In contrast, the median Geneva scale score was significantly higher in the PE group than in the non-PE group (6 (6–8) vs. 4 (2.5–6); *p* = 0.005).

To identify a good discrimination of PE in our elderly population, the cutoff point that maximized sensitivity and specificity on the Wells scale was 2.5 and that on the Geneva scale was 5. After taking these values into account, in older people infected with COVID-19, the positive predictive value (PPV) of the Wells scale was similar to that of the Geneva scale (55% vs. 51%). Regarding diagnostic accuracy, the sensitivity of the Geneva scale performed better than the Wells score in the probability of detecting PE, and the area under the curve (AUC) for the Geneva scale was better (Figure 3a) than that for the Wells score (Figure 3b).

### 3.3. DD and Clinical Score for the Geriatric Population with PE and COVID-19

A cutoff value of 4.3 mg/L in patients over 75 years of age was calculated to categorize the patients with and without PE, and this optimal value showed an intermediate sensitivity of 52.9% and a high specificity of 93.9% (80.4–98.3%) (Figure 4). The cutoff point adjustment for the DD level from 1.0 to 4.3 mg/L resulted in a significant rise in the specificity (30.3% to 93.9%) but reduced the sensitivity (Table 3). Moreover, the combination of a 4.3 mg/L DD cutoff point with several variables to determine the pretest probability of PE, such as the value of 2.5 on the Wells scale and 5 as intermediate risk on the Geneva scale indicating intermediate risk (both obtained by ROC curves), led to improvements in the PPV and the negative predictive value (NPV) (Table 4). 

## 4. Discussion

We demonstrated a novel DD cutoff value for the oldest COVID-19 patients to rule out PE. To date, no study has been carried out in extremely geriatric patients to improve the diagnosis of PE in this population. For the first time, we showed that increased DD values and clinical rules can help improve the diagnosis of PE in this extremely old population.

The diagnosis of PE in elderly patients with COVID-19 is challenging, and several situations complicate the diagnosis: (i) age, (ii) the initiation of anticoagulant treatment, (iii) infection by SARS-CoV-2 that causes a prothrombotic effect with a high incidence of embolic phenomena and excessive DD levels and (iv) the lack of symptoms. Therefore, the usefulness of this isolated biomarker with a new cutoff point can be very helpful in clinical practice for the initial diagnosis of PE [23].

The combination of clinical criteria and DD to determine the probability of PE in a general population using several predictive rules and a scheme of three categories based on the clinical probability has been previously studied, but this kind of study has not been performed in a COVID-19 cohort [12,13]. In COVID-19, a hypercoagulable status with impaired DD levels affects the pretest probability diagnosis, making this algorithm difficult to use for follow-up. CTPA is the gold standard [24] technique for diagnosing PE, but during the first stage of the COVID-19 pandemic, its use was limited by several factors, such as the isolation of patients, the risk of viral spread and the significant number of hospitalized patients [25]. 

Thus, elevated DD levels are a sign of excessive activation of coagulation and hyperfibrinolysis [26]. In our cohort, DD levels over 4.33 mg/L to detect PE in elderly COVID-19 patients showed a sensitivity of 52.9%. Compared with other series of non-COVID-19 patients, this sensitivity is low, and when the DD value is negative, the sensitivity of this biomarker is over 95%, especially in low- and intermediate-risk patients [27,28,29]. However, it must be emphasized that 96% of our population was elderly, had COVID-19 and had previous treatment with heparin, and there is a decrease in the DD value of approximately 25% within the first 24 h after starting heparin. All these factors may have influenced the lower sensitivity (from 95.5% to 89.4%), as has been shown [30]. However, the specificity of DD in our cohort was high (93.9%), with a PPV and NPV that were not negligible (81.8% and 79.5%, respectively). It is known that the specificity of DD decreases as the age of the patient increases; however, when the cutoff point is adjusted, this specificity increases, as has been shown in other studies in non-COVID patients [31,32]. An increased DD cutoff value will increase the specificity, which will reduce the number of false positives. This may be crucial in clinical practice in two important aspects: (i) when referring a frail, elderly patient with multiple comorbidities to receive CTPA, which would reduce the risk of contrast nephropathy and contagion as well as the medical costs during the pandemic, and (ii) these preliminary data may suggest the necessity for early anticoagulant treatment in the appropriate patients.

Similar studies in non-COVID-19 populations have shown the importance of increasing the DD value to minimize unnecessary examinations without influencing the efficacy of DD to rule out PE [33]. Likewise, according to our results, a low sensitivity of 65%, specificity of 96.7%, PPV of 86% and NPV of 89.4% have been reported in COVID-19 patients with DDs above 3.5 mg/L [34], but when the cutoff point is increased to >5 mg/L, the predictive value is 75.5% [35].

There is much controversy regarding standard prophylaxis or anticoagulation treatment in hospitalized patients with COVID-19. Today, anticoagulants probably reduce thromboembolic events compared to prophylactic doses, but at the expense of a higher risk of bleeding. An isolated DD would not be an indicator for anticoagulation if the patient is at low or moderate risk. However, if this value added to a high preclinical risk of PE and if imaging techniques were not available, the patient would benefit from anticoagulant treatment. It is important to take into account the embolic risk vs. the hemorrhagic risk, since we are facing a very old, frail population with many comorbidities. To date, there are no validated venous and hemorrhagic embolic risk scales in COVID-19.

Some studies have demonstrated the usefulness of both scales and have confirmed the diagnostic performance of the Wells and Geneva scales [14,15] to estimate the pretest probability [36]. There was no significant difference between the Wells and Geneva rules when different pretest probability scales for PE in elderly COVID-19 patients were compared, although the Geneva scale had a greater sensitivity (82.4% vs. 64.7%) with an NPV of 87%.

Chagnon et al. [37] described a similar proportion of patients with low, intermediate and high risks based on the pretest probability for PE and by using the same two scales in patients from the emergency room. They concluded that the Geneva rule was more reliable when the clinical data were added. In contrast, our results showed a higher proportion of patients with moderate risk on the Geneva scale than on the Wells scale (82.2% vs. 58.8%). In the revised Geneva score, the high prevalence of patients in the intermediate probability range may be related to novel items, including age (all of our patients were older than 75 years) and heart rate of 75–94 beats per minute, which represents 3 points. Our cutoff was an average HR of 88 bpm, unlike the Wells scale, which values HR > 110 bpm as an item.

In our study cohort, the superiority of the modified Geneva scale in elderly patients with COVID-19 was proven, which is in disagreement with studies that evaluated elderly non-COVID-19 patients. Furthermore, Di Marca et al. reported that the Wells scale demonstrated a better discrimination index than the Geneva scale [7]. However, their study is not comparable to ours due to the features of the population recruited and the study methodology, where only patients in the high-probability risk group were referred for a CTPA. If the group had a moderate risk, the DD value was assessed and was only considered elevated when it was greater than 0.5 mg/L, as recommended by the European Society of Cardiology guidelines.

A meta-analysis [38] concluded that when the cutoff point of the Wells scale is less than 2, its sensitivity is greater, and when the cutoff point is 4, the sensitivity is reduced to 60% with a high specificity. Thus, the lower the clinical probability, the higher the Wells score sensitivity. These data are in line with our findings: the cutoff point of the Wells rule was over 2.5, and the specificity increased to exclude low- and moderate-risk cases, which influenced the sensitivity. Thus, intermediate case percentages are a flaw for these scales and need to be supported by an increased DD level. Clinical rules provide a tough basis that, in addition to DD, turn out to be a multiparametric tool that tailors the decision to perform a CTPA. According to a recently published study that evaluated COVID-19 patients [39], the different DD cutoff points associated with the Wells scale were valued over 2 points, demonstrating that the specificity of the Wells scale is higher when a DD cutoff point over 3000 ng/mL is added, at the cost of a low sensitivity of 57.1%.

Current guidelines show that a high pretest probability leads to an excessive use of CTPA, and using a more specific Wells score could safely reduce the number of unnecessary invasive tests. Herein, we proposed a combination of the Wells and Geneva rules that are associated with increased DD levels to accurately diagnose PE in an elderly COVID-19 population, which shows an improvement in the specificity of both rules to 96.8% and 93.9%, respectively. This clinical strategy was based on the association of a DD cutoff point with a Wells scale score of less than 2.5 and a Geneva scale score of less than 4. For the first time, we have presented a novel and accessible strategy to avoid unnecessary invasive tests in elderly COVID-19 patients to improve the diagnosis of PE.

Our results raised several questions about the diagnosis of PE in older patients with COVID-19, such as which is the best DD value for a clinical suspicion of PE, and what is the best pretest patient assessment that may avoid unnecessary tests. Therefore, in light of the results obtained, new algorithms and multiparametric clinical tools should be proposed for this fragile population.

## 5. Limitations

Our work has some limitations. First, there is a lack of scientific literature on COVID-19 in the elderly population and the associated biomarkers. In patients infected by COVID-19, most studies carried out are retrospective; therefore, it is important to validate a displaced cutoff point combined with age and COVID-19 infection. Second, confounding biases, including the clinical diagnosis, and limited knowledge of the pathophysiology and biomarkers in COVID-19 patients, need to be supported by future multicenter studies. Third, the incidence of PE could have been underestimated at the first stage of the pandemic due to the lower number of patients referred for CTPA, mainly because of the critical situation of elderly patients. Finally, the dynamic changes in the DD levels from admission to discharge and the low experience with the use of this biomarker in COVID-19 patients could have been influenced by the age of our cohort. 

## 6. Conclusions

We proposed a novel, noninvasive approach to the elderly COVID-19 population to rule out PE. We demonstrated that the modified Geneva scale was more accurate than the Wells scale for classifying patients with suspected PE. A DD cutoff point of >4.33 mg/L has a high specificity and a high NPV for discriminating false positives. The combination of the DD level over 4.33 mg/L and the clinical probability scales increases the specificity and NPV, leading to reduced exposure to unnecessary tests in this fragile population that commonly has comorbidities.

## Figures and Tables

**Figure 1 jcm-10-05433-f001:**
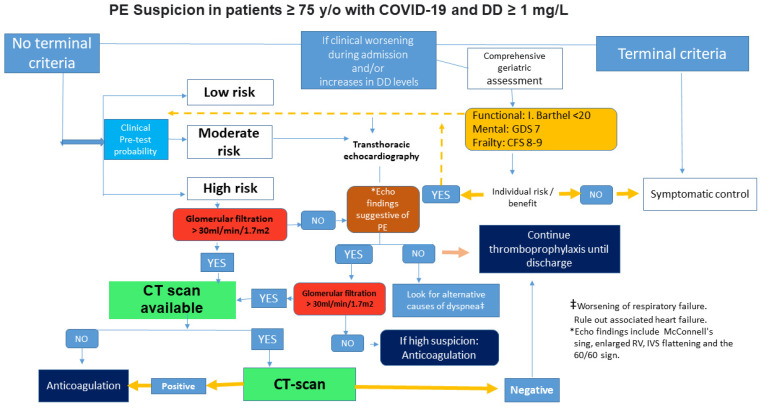
Algorithm for pulmonary embolism in elderly patients with COVID-19. High-risk patients were considered those who scored >6 on the Wells scale or >11 on the revised Geneva scale, and those with D-dimer ≥ 1 mg/L with a torpid evolution; RV—Right ventricle; IVS—Interventricular septum; GDS—Global deterioration scale; CFS—Clinical frailty scale.

**Figure 2 jcm-10-05433-f002:**
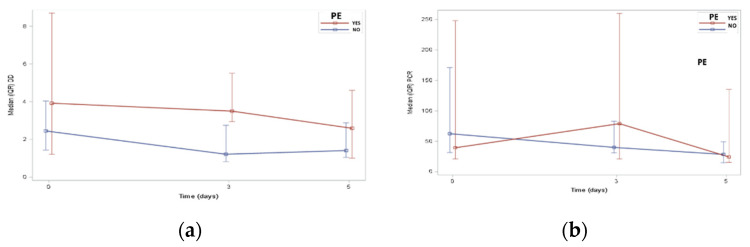
Schemes following biomarker evolution during hospitalization: (**a**) D-dimer evolution curve according to groups, and (**b**) C-reactive protein evolution curve according to groups.

**Figure 3 jcm-10-05433-f003:**
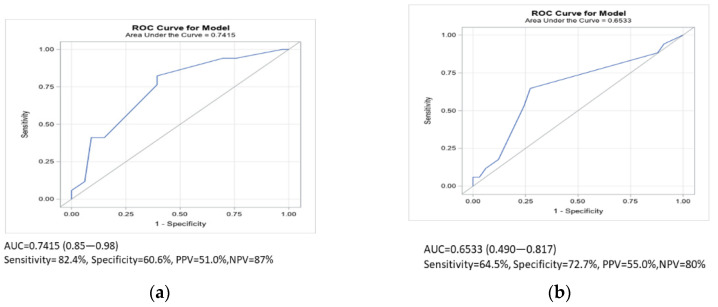
Predictive values of the Wells score and the revised Geneva score: (**a**) ROC curve scale modified Geneva and (**b**) ROC curve scale Wells.

**Figure 4 jcm-10-05433-f004:**
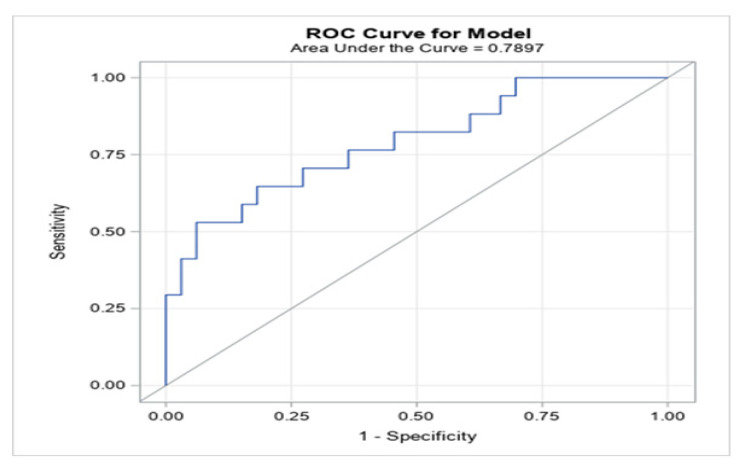
Receiver operating characteristic curve for D-dimer levels as a parameter for predicting pulmonary embolism in the COVID-19 cohort. The AUC for the model was 0.7897 (0.652–0.927). The threshold for the D-dimer level was 4.3 mg/L, which showed a sensitivity of 52.9%, specificity of 93.9%, positive predictive value of 81.8% and negative predictive value of 79.5%.

**Table 1 jcm-10-05433-t001:** Characteristics of the population.

Features	Global Population	PE (*n* = 17)	Non-PE (*n* = 33)	*p*
Age, years	85.5 (80–90)	83 (80–86)	88 (81–91)	0.264
Sex (% male)	26 (52.0)	9 (52.9)	17 (51.5)	0.924
**Place of origin (%)**				0.475
Home	11 (22.0)	5 (29.4)	6 (18.2)	
Nursing home	39 (78.0)	12 (70.6)	27 (81.8)	
**BMI (*n* = 49)**				**0.046**
Low weight	5 (10.2)	1 (5.9)	4 (12.5)	
Normal weight	25 (51.0)	7 (41.2)	18 (56.3)	
Overweight	8 (16.3)	2 (11.8)	6 (18.8)	
Obesity	11 (22.4)	7 (41.2)	4 (12.5)	
Time from clinical symptoms to admission, days	8 (5–10)	7 (4.5–9)	8 (5–10)	0.403
Time from COVID diagnosis to CT scan, days	14 (8–23)	15 (10–23)	12 (8–22)	0.362
Days of hospitalization	14.5 (11–21)	15 (13–28)	14 (10–20)	0.246
**Comorbidities, *n* (%)**				
Oncological history	10 (20.0)	6 (35.3)	4 (12.1)	0.070
DVT	1 (2.0)	1 (5.9)	0 (0)	0.340
PE	3 (6.0)	0 (0)	3 (9.1)	0.542
Trauma	1 (2.0)	0 (0)	1 (3.0)	1.000
Neoplasia in palliative treatment	2 (4.0)	1 (5.9)	1 (3.0)	1.000
Lower limbs pain	2 (4.0)	1 (5.9)	1 (3.0)	1.000
**PE symptoms**				
Heart rate, beats/min (range)	88 (80–100)	96 (86–109)	86 (76–96)	**0.015**
Tachycardia classification				**0.013**
75–94	22 (44.0)	5 (29.4)	17 (51.5)	
>94	20 (40.0)	11 (64.7)	9 (27.3)	
DVT signs	3 (6.0)	2 (11.8)	1 (3.0)	0.264
New DVT	3 (6.0)	2 (11.8)	1 (3.0)	0.264
Pain/edema lower limbs	4 (8.0)	3 (17.6)	1 (3.0)	0.108
Arterial embolic event				0.108
Lower limb ischemic events	2 (4.0)	1 (5.9)	1 (3.0)	
Embolic stroke	2 (4.0)	2 (11.8)	0 (0)	
Severity of the disease: CURB65	3 (2-3)	2 (2-3)	3 (2-3)	0.431
**Geriatric assessment**				
Dependency	35 (70.0)	10 (58.8)	25 (75.8)	0.216
Frailty	32 (64.0)	10 (58.8)	22 (66.7)	0.584
Polypharmacy	34 (68.0)	11 (64.7)	23 (69.7)	0.720
Dementia	20 (40.0)	6 (35.3)	14 (42.4)	0.626
**Symptoms at hospitalization**				
Fever	22 (44.0)	6 (35.3)	16 (48.5)	0.373
Falls	9 (18.0)	5 (29.4)	4 (12.1)	0.242
Dyspnea	41 (82.0)	13 (76.5)	28 (84.8)	0.468
Loss of appetite	12 (24.0)	2 (11.8)	10 (30.3)	0.181
Asthenia	18 (36.0)	3 (17.6)	15 (45.5)	0.052
Delirium	13 (26.0)	3 (17.6)	10 (30.3)	0.499
Cough	11 (22.0)	3 (17.6)	8 (24.2)	0.728
**Pneumonia**				1.000
Unilateral	12 (26.7)	4 (25.0)	8 (27.6)	
Bilateral	33 (73.3)	12 (75.0)	21 (72.4)	
**Medication**				
Hydroxychloroquine	37 (75.5)	12 (75.0)	25 (75.8)	1.000
Azithromycin	27 (55.1)	8 (50)	19 (57.6)	0.617
Steroids	24 (49)	8 (50)	16 (48.5)	0.921
**PE prophylaxis**	47 (94.0)	16 (94.1)	31 (93.9)	1.000
Type of anticoagulation				0.725
Prophylactic dose	35 (70.0)	11 (64.7)	24 (72.7)	
Full anticoagulation	12 (24.0)	5 (29.4)	7 (21.2)	
Time of prophylaxis	10 (8-14)	10 (9-13)	12 (6-15)	0.623
**Mortality**	10 (20.0)	3 (17.6)	7 (21.2)	1.000

BMI—body mass index; CURB-65—severity score for predicting mortality from community-acquired pneumonia; DVT—deep vein thrombosis; NOACS—non-vitamin K antagonist oral anticoagulants; PE—pulmonary embolism. Data are presented as the medians and interquartile ranges or numbers (%). Bold indicates statistically significant variables (*p* < 0.05).

**Table 2 jcm-10-05433-t002:** Laboratory characteristics between the PE and non-PE groups.

Characteristics	PE (*n* = 17)	Non-PE (*n* = 33)	*p*-Value
D-Dimer mg/L	4.33 (2.40–7.17)	1.39 (1.01–2.75)	**<0.001**
NT-Pro-BNP pg/mL	1273 (444–1908)	1003 (501–2240)	0.946
Troponin ng/L	40 (40–53)	40 (40–55)	ND
CRP mg/L	39.4 (21.0–248.0)	62.5 (31.6–170.9)	0.802
Ferritin	225 (159–463)	243 (185–737)	0.316
Lymphocytes	0.72 (0.55–1.20)	0.75 (0.40–1.06)	0.630

PE—pulmonary embolism; NT-Pro-BNP—N-terminal pro-brain natriuretic peptide; CRP—C-reactive protein. Bold indicates statistically significant variables (*p* < 0.05).

**Table 3 jcm-10-05433-t003:** D-Dimer values in the geriatric population with PE and COVID-19.

DD mg/L	Sensitivity	Specificity	PPV	NPV	False Positives
1.0	100%	30.3%	42.5%	100%	23%
1.5	82.4%	54.5%	48.3%	85.7%	15%
2.0	76.5%	60.6%	50%	83.3%	13%
2.5	70.6%	69.7%	54.5%	82.1%	10%
3.0	64.7%	78.8%	61.1%	81.3%	7%
3.5	58.8%	81.8%	62.5%	79.4%	6%
4.33	52.9%	93.9%	81.8%	79.5%	2%

DD—D-dimer; PPV—positive predictive value; NPV—negative predictive value.

**Table 4 jcm-10-05433-t004:** Predictive values of the DD levels, Wells score, revised Geneva score and their combination to detect pulmonary embolism in elderly patients with COVID-19.

Items	S	E	PPV	NPV

Wells score	64.5	72.7	55.0	80.0
Revised Geneva score	82.4	60.6	51.0	87.0
D-dimer	52.9	93.9	81.8	79.5
Wells score with D-dimer	35.3	96.8	85.7	74.4
Geneva score with D-dimer	47.1	93.9	80	77.5

S—sensitivity; E—specificity; PPV—positive predictive value; NPV—negative predictive value.

## Data Availability

The data that support the findings of this study are available upon request from the corresponding author. The data are not publicly available due to privacy or ethical restrictions.

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
