# Peer review of "Elderly Population with COVID-19 and the Accuracy of Clinical Scales and D-Dimer for Pulmonary Embolism: The OCTA-COVID Study"

_jcm, 2021, doi:10.3390/jcm10225433_

Round 1

Reviewer 1 Report

  1. Pg 1 Line 46, should state SARS-COV2
  2. Figure 1 - more explanation should be given that this is quite a complex diagram in identifying high risk patients
  3. What proportion of symptomatic patients ended up getting scan using your pre-clinical algorithm? Do you think the fact that not all had a CTPA - was this a confounding factor?
  4. Pg Line 100 - was only tachycardia data collected?
  5. Pg 3 line 114 - should define the machine/technology used for D-dimer testing
  6. Table 1 - highlight statistically significant results
  7. Table 2 - troponinas should be troponin. Please check your terminology to ensure appropriate English
  8. Resolution of figure 2 should be looked into
  9. Your suggested D-dimer of 4.33 - is that taken at the start of admission or during the course of admission?
  10. Would D-dimer at D3 be a better discriminator?
  11. Figure 4 D-dimer not D-dimero
  12. Ensure decimal point used for numbers e.g 4.3 instead of 4,3
  13. Discussion paragraph 1 should be re-worded. The has been studies evaluating D-dimer in COVID but your study is limited to the extreme elderly
  14. Given the latest articles suggesting a D-dimer >4 should warrant therapeutic anticoagulation, do you think this is related to underlying PE ?Comment on this in your paper
  15. Limitations should be before conclusion

Author Response

Q1:  Pg 1 Line 46, should state SARS-COV2

A1: Thank you very much for your comment. We have made this correction in the text.

Q2:Figure 1 - more explanation should be given that this is quite a complex diagram in identifying high risk patients.

A2: We agree with your suggestion. We have included this information in the manuscript, below Figure 1.

“High risk patients were considered those who scored > 6 on the Wells scale or > 11 on the revised Geneva scale, and those with D-dimer ≥ 1 mg/l with a torpid evolution.”

Q3:  What proportion of symptomatic patients ended up getting scan using your pre-clinical algorithm? Do you think the fact that not all had a CTPA - was this a confounding factor?

A3: The fact that not all patients had a CTPA is probably a confounding factor, as the reviewer commented. During March, when the data were collected, the mortality rate was high compared to the percentage of pulmonary embolism (PE) detected. It is possible that in this phase, new cases of PE were lost. However, when we started using the algorithm, we observed that the events were increasing. It should be emphasized that it was difficult to make this selection, as this was a very old, fragile population that had to be previously well evaluated by the geriatrician to define whether patients met terminal criteria and whether they would benefit from anticoagulation. Therefore, the number of symptomatic patients that we considered candidates for CTPA was 50.

Q4: Pg Line 100 - was only tachycardia data collected?

A4: The overall heart rate of the sample was collected and classified as seen in Table 1 (between 75-94 bpm and > 94 bpm), following the indications of the revised Geneva scale. Other clinical signs that were assessed included breathing rate, oxygen saturation, and blood pressure. The median breathing rate was 20 breaths per minute (IQR 18-28), and the median saturation was 93% (IQR 88-95%). The median blood pressure was 129 mmHg (IQR 110-148). These data were not included to simplify and improve Table 1 understanding.

Q5: Pg 3 line 114 - should define the machine/technology used for D-dimer testing

A5: Thank you very much for your suggestion. We have included this information in the manuscript (lines 116-119).

“The determination of plasmatic DD was carried out using the VIDAS® DD technique (bioMérieux, Lyon, France), a sandwich-type immunoenzymatic method, in 2 stages, with a final detection by enzyme-linked fluorescence assay (ELFA). The concentration was expressed in micrograms per milliliter of fibrinogen equivalent units.”

Q6: Table 1 - highlight statistically significant results

A6: We have highlighted the significant results

Q7  Table 2 - troponinas should be troponin. Please check your terminology to ensure appropriate English

A7: Thank you very much for your comment. We have made this correction.

Q8  Resolution of figure 2 should be looked into

A8: Thank you very much for your comment. We have increased the resolution of this image.

Q9  Your suggested D-dimer of 4.33 - is that taken at the start of admission or during the course of admission?

A9: We have clarified this question in the text. As mentioned in the methodology, the D-dimer (DD) value was adjusted based on the patient's age following the current PE guidelines and was considered elevated when it was above 1 mg/l (lines 126-128). We used the peak DD value from either the beginning of admission or during the course of admission. We consider that when the DD value is > 4.3 mg/l, either at the time of admission or throughout hospitalization, the suspicion of PE should be high.

We have included this information in the manuscript (line 120-121).

“We included the peak DD value of D-dimer, either from the beginning of admission or if there was a non-COVID related clinical event  during the course of the hospitalization”

Q10  Would D-dimer at D3 be a better discriminator?

A10: In Cui's study, a DD value of 3 mg/l was used as the cutoff point, with a sensitivity, specificity, and negative predictive value of 76.9%, 94.9%, and 92.5%, respectively. We agree that this value could also be considered a good discriminator of embolic events; however, we believe that different populations must be individualized and nuanced. In Cui's study, the population was younger, with a mean age of 68 years. In our case, we had a very elderly population with high levels of DD according to their age. Furthermore, 70% of the patients in our study were previously treated with heparin prophylaxis, and 24% were treated with full doses of anticoagulation, which is associated with a decrease in the DD values in blood. In our analysis, a DD value of 3 mg/l did not obtain an adequate area under the receiver operating characteristic (ROC) curve.

Q11  Figure 4 D-dimer not D-dimero

A11: Thank you very much for your support. We have made this correction.

Q12  Ensure decimal point used for numbers e.g 4.3 instead of 4,3

A12: We have checked the decimal points and corrected those that were wrong.

Q13  Discussion paragraph 1 should be re-worded. The has been studies evaluating D-dimer in COVID but your study is limited to the extreme elderly

A13: We completely agree with your suggestion. We have modified this paragraph in the manuscript (lines 233-236).

“To date, no study has been carried out in extremely geriatric patients to improve the diagnosis of PE in this population. For the first time, we have showed that increased DD values and clinical rules can help improve the diagnosis of PE in this extremely old population”.

Q14  Given the latest articles suggesting a D-dimer >4 should warrant therapeutic anticoagulation, do you think this is related to underlying PE ?Comment on this in your paper

A14: We completely agree with your comment. We have included an explanatory paragraph in the manuscript (lines 277-285).

“There is much controversy regarding standard prophylaxis or anticoagulation treatment in hospitalized patients with COVID-19. Today, anticoagulants probably reduce thromboembolic events compared to prophylactic doses but at the expense of a higher risk of bleeding. An isolated DD would not be an indicator for anticoagulation if the patient is at low or moderate risk. However, if this value added to a high preclinical risk of PE and if imaging techniques were not available, the patient would benefit from anticoagulant treatment. It is important to take into account the embolic risk versus the hemorrhagic risk, since we are facing a very old, frail population with many comorbidities. To date, there are no validated venous or hemorrhagic embolic risk scales in COVID-19.”

Q15  Limitations should be before conclusion

A15: Following your recommendation, we have changed the order of the limitations and conclusions.

Reviewer 2 Report

Manuscript ID: jcm-1436561

Title: Oldest-old population with COVID-19 and the accuracy of clinical scales and D-dimer for pulmonary embolism

Authors: Maribel Quezada-Feijoó, Mónica Ramos, Isabel Lozano-Montoya, Mónica Sarró, Verónica Cabo Muiños, Rocío Ayala, Francisco Javier, Gómez-Pavón, Rocío Toro Submitted to section: Vascular Medicine,

The authors examined data of Patients over 75 years of age hospitalized with COVID-19 with a clinical suspicion of PE  from the Acute Geriatrics Unit between March and May 2020, in order to define the diagnostic accuracy and reproducibility of the clinical probability scales, Wells and Geneva, and their association with the D-dimer (DD) in the diagnosis of PE in elderly patients with COVID-19. The study results proposed a new approach for elderly patients with COVID-19 to rule out PE. A DD cutoff point of > 4.33 mg/l was proposed. The sum of DD, over 4.33 mg/l, and the clinical probability scales increases the specificity and the negative predictive values, so that the explosion to CT scan in  fragile population could be avoided.

The study is an interesting research and appears in line with the aims and scope of the Journal. The study methodologies are in agreement with the highest ethical standards. The manuscript is written in standard English.

However, despite the article is of interest, I have pointed out some criticisms that in my opinion significantly impair the study relevance.

  • The study population comes from the OCTA-COVID cohort. Among them 50 had a clinical suspect for PE. If the inclusion criteria are well defined, the definition of the exclusion criteria are lacking.
  • In figure 1, terminal criteria are named but not defined in the text.
  • In the results, a cutoff value for the Wells scale and Geneva scales was chosen. I suspect that the better value was chosen based on the results of the ROC curves, but it’s not specified.
  • Even if accurate, table 1 is a little bit confusing and not handy to read. The elements must be separated in some way: indentation, bulleted lists…

Author Response

Q1:The study population comes from the OCTA-COVID cohort. Among them 50 had a clinical suspect for PE. If the inclusion criteria are well defined, the definition of the exclusion criteria are lacking.

A1: Thank you very much for your suggestion. We have defined the exclusion criteria in the manuscript (lines 71-74).

 “Patients under 75 years of age, those with palliative needs, those diagnosed by the attending team, and those who did not meet the diagnostic criteria for COVID-19 were excluded. Patients with a high suspicion of PE who could not undergo a computed tomography (CT) scan and those who declined to participate were also excluded.”.

Q2: In figure 1, terminal criteria are named but not defined in the text.

A2: We have included this information in the text (lines 90-93).

“We considered terminal criteria as advanced organ disease, such as severe lung failure with irreversible damage, end-stage heart disease, advanced kidney disease without dialysis criteria or advanced treatment subsidiary among others, according to The National Hospice Organization.”

  • Medical guidelines for determining prognosis in selected non-cancer diseases. The National Hospice Organization. Hosp J. 1996;11(2):47-63. doi: 10.1080/0742-969x.1996.11882820. PMID: 8949013).

Q3: In the results, a cutoff value for the Wells scale and Geneva scales was chosen. I suspect that the better value was chosen based on the results of the ROC curves, but it’s not specified.

A3: You are absolutely correct. To identify a good discrimination of PE in our older population, we used the cutoff point that maximized the sensitivity and specificity on the Wells scale, which was 2.5, and that on the Geneva scale, which was 5.

We have modified the text for a better understanding (lines 219-223).

“Moreover, the combination of a 4.3 mg/dl DD cutoff point with several variables to determine the pretest probability of PE, such as the value of 2.5 on the Wells scale and 5 on the Geneva scale indicating intermediate risk (both obtained by ROC curves), led to improvements in the PPV and the negative predictive value (NPV) (Table 4).”

Q4: Even if accurate, table 1 is a little bit confusing and not handy to read. The elements must be separated in some way: indentation, bulleted lists…

A4: We completely agree with your comment. We have highlighted and underlined the sections for better visualization and we have highlighted the significant results.